# Recent Progress of MIL MOF Materials in Degradation of Organic Pollutants by Fenton Reaction

**Keru Xiao [1], Bao Shu [2], Kangle Lv [1], Peipei Huang [1], Qing Chang [1,\*], Laiyan Wu [1], Songbo Wang [1] and Lingling Cao [1]**

[1] Key Laboratory of Catalysis and Materials Science of Hubei Province, College of Resources and Environmental Science, South-Central Minzu University, Wuhan 430074, China

[2] Guangdong Dongguan Ecological and Environmental Monitoring Station, Dongguan 523000, China

\* Correspondence: changqinghust@163.com

**Abstract:** In recent years, environmental pollution has become more serious, especially the organic pollutants. Metal organic frameworks (MOFs) are promising materials used to degrade pollutants recently. Among them, Materials Institute Lavoisier frameworks (MILs) have been widely engaged due to their good stability and unique structural characteristics. This paper systematically analyses and summarizes the progress of MILs in degradation of organic pollutant by Fenton reaction in recent years. The MILs, especially four types of MILs, including MIL-100, MIL-101, MIL-88, and MIL-53, are first described and classified. Then, the common synthesis methods (hydrothermal synthesis, steam-assisted synthesis, and microwave-assisted synthesis) of MIL are summarized and compared. Modification and activation of MILs to obtain good degradation effect are also introduced and discussed. Finally, the applications of MILs in Fenton reaction are reviewed and their future development is prospected.

**Keywords:** metal organic frameworks; MILs; Fenton reaction; degradation





## 1. Introduction

With population growth, economic development, and the growth in the living standard, the water pollution caused by the discharge of industrial, domestic, and agricultural wastewater has seriously endangered the health of human life [1]. Every year, hundreds of millions of tons of industrial wastewater are discharged into rivers. The most threatening are persistent organic pollutants (POPs), which are not subject to natural degradation [2,3]. Due to their special properties, POPs can migrate to distant locations and have high stability [4]. Traditional sewage treatment plants cannot effectively remove them using traditional treatment methods. Overusing POPs and slow decomposition will cause their residues to flow into the aquatic environment [5]. After entering the human system, POPs can bioaccumulate and harm human health, posing varying degrees of threat to cardiovascular, endocrine, nervous, and immune systems [6]. Therefore, in addition to preventing water pollution, it is also crucial to research sufficient and effective water treatment methods to achieve the effect of removing these organic pollutants.

Advanced oxidation processes (AOPs) have attracted wide attention over the past two decades as the effective chemical oxidation process for organic pollutants. The Fenton oxidation process has been an extremely significant research area due to its mild reaction conditions, simple operation, and easy control [7]. It is the process by which $H_2O_2$ and $Fe^{2+}$ convert an organic compound into an inorganic compound. As the main active substance, hydroxyl radicals ($\bullet OH$) are vital in degradation. Usually, $Fe^{2+}/Fe^{3+}$ coexist in the Fenton system and react with $H_2O_2$, both of which are occurring simultaneously (Equations (1)

and (2)). It is not difficult to see that their redox cycle efficiency largely determines the reaction system's degradation efficiency.

$$Fe^{2+} + H_2O_2 \rightarrow Fe^{3+} + OH^- + \bullet OH \tag{1}$$

$$Fe^{3+} + H_2O_2 \rightarrow Fe^{2+} + H^+ + HO_2{}^- \tag{2}$$

For the degradation of some organic pollutants, many researchers have valued MOFs due to their uniform distribution of elements, ultra-high surface sensitivity, simple synthesis, and abundant porous structure [8]. These advantages lead to MOFs being more widely used in the degradation of organic pollutants in recent years instead of homogeneous catalysts. Homogeneous catalysts, especially, have disadvantages, such as a narrow operating pH range and easy-to-produce sludge. MILs are special MOFs that can use trivalent transition metal ions (such as Fe, Al, and Cr) to coordinate with carboxylic acid ligands [9]. MILs have a very high surface area and can be used to adsorb and remove organic contaminants from water [10]. Wang et al. synthesized $NH_2$-MIL-53(Fe) and confirmed its strong adsorption of Congo red [11]. Tran et al. successfully absorbed and degraded hazardous pollutants by utilizing the respiration of $NH_2$-MIL-53(Al) materials [12]. Xiao et al. prepared MIL-101(Fe, Co) for the degradation of Rhodamine B in the water, and the degradation efficiency was achieved at 99% within 15 min [13]. Moreover, Tan et al. also found that MIL-88A had a strong adsorption and degradation effect on specific dyes under visible light irradiation [14]. MILs are also widely used in gas storage and separation, drug transportation, etc.

Approximately 300 journal articles have been published on the application of MILs in Fenton catalysis, as presented in Figure 1. As far as we know, the synthesis of MILs, their modification and activation, and their applications in different Fenton reactions have not been summarized at the forefront, so it is important to summarize and prospect the Fenton-based degradation systems involved in MILs. In this review, we summarize the classification and research progress of MILs as catalysts in Fenton reactions and systematically describe the use and application of MILs in many ways. At the same time, the processing and modification of basic materials in MILs are extracted, including (1) metal sites, (2) ligands, and (3) activation. Ultimately, we express our vision for the research and application of MILs in the Fenton, electro-Fenton, and photo-Fenton reaction systems.

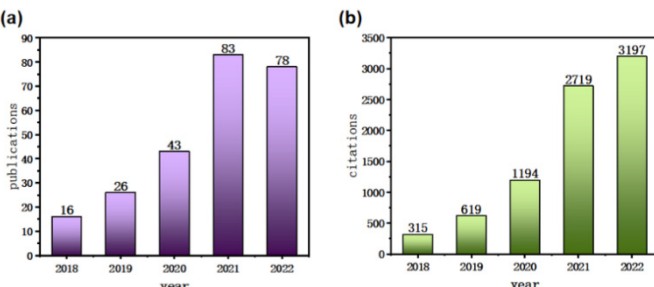

**Figure 1.** Number of (**a**) publications and (**b**) citations obtained from a survey performed in the Web of Science using "MIL" AND "Fenton" as keywords (data from 2018 to 2022).

## 2. Research Progress of MILs as Catalysts in Fenton Reaction

Hydrogen peroxide ($H_2O_2$) and divalent iron ions ($Fe^{2+}$) are used to convert organic matter into an inorganic state in a Fenton reaction. Fenton reaction has a greater ability to remove organic pollutants than others. Therefore, it is naturally widely used in the treatment of industrial wastewater and recognized by the outside world [7].

At present, there are many studies on using MILs in Fenton-like reactions to catalyze degradation of organic pollutants in water [15]. Among all MOF species, they are highly stable and have many applications in catalysis due to their unique pore structures.

### 2.1. The Development of MILs

MOFs are new kinds of porous materials, which are generally coordinated polymer composed of the spatial extension of organic ligands with metal ions as the connection points [16]. They have the advantages of organic–inorganic mixture, ultra-high surface sensitivity, large specific surface area, highly adjustable porosity, topology diversity, and tailoring [17]. Because of the remarkable structural properties, they have been widely used in various technical and industrial fields. However, since the reaction environments faced by MOFs often require exposure to high temperatures, chemicals, and water [18], it is very important to select a MOF with strong stability and unique properties to remove multiple pollutants exposed to harsh environments.

MILs were first discovered by Professor Férey of the University of Versailles in France and his research group. MILs have received wide attention because of their huge specific surface area and stable structural peculiarity, as well as their amazing applications in catalysis [10]. It is worth emphasizing that detailed degradation process of organic pollutants by MILs is involved mainly with the Fenton reaction, which lays a solid theoretical foundation for the development and research of MILs. Furthermore, it inspires people to improve the experimental ideas of new materials [15].

### 2.2. The Classification of MILs

There are many types of MILs which have been widely applied in catalysis, such as MIL-125, MIL-101, MIL-100, MIL-88, MIL-68, and MIL-53. In this study, we focused on four types of MILs, including MIL-100, MIL-101, MIL-88, and MIL-53, combined with their structural characteristics and unique characteristics. We also explained the similarities and differences among different materials of the same type of MILs.

#### 2.2.1. MIL-100

The MIL-100(Cr) comprises chromium-trimers and carboxylate moieties. The large Langmuir surface area (3100 m$^2$ g$^{-1}$) and high thermal stability (270 °C) of MIL-100(Cr) make it a promising catalytic material. Later, MIL-100(Fe) was synthesized under hydrothermal conditions [19]. It is a highly stable material with a three-dimensional porous structure (Figure 2a), including two groups of mesoporous cages [20]. It has many Lewis active sites and plays a key role in adsorption reactions. At the same time, it has a large Langmuir surface area (2800 m$^2$ g$^{-1}$), endowing it with more active sites which can directly interact with pollutants. MIL-100 has not only good structural stability and a large specific surface area but also good applications in the field of optical catalysis.

#### 2.2.2. MIL-101

The MIL-101(Cr) is composed of octahedral clusters of trimeric chromium(III) interlinked via 1,4-benzenedicarboxylates [21]. It has a very high Langmuir surface area of 5900 m$^2$ g$^{-1}$ and thermal stability up to 250 °C (Figure 2b) [22]. Its large Langmuir surface area enables it to adsorb pollutants in water. Researchers have been keen to improve the material to increase its specific surface area to show greater advantages in the adsorption field. MIL-101(Fe) has a highly active iron center and low toxicity, with thermal stability up to 300 °C [23]. Later, the amination of MIL-101 materials began to appear, which greatly improved its catalytic performance. For example, NH$_2$-MIL-101(Al/Fe) can degrade organic pollutants more effectively than can MIL-101(Al/Fe).

#### 2.2.3. MIL-53

The MIL-53(Al) is created by connecting octahedra AlO$_4$(OH)$_2$ immense trans corner-sharing across their (BDC) 1,4-benzenedicarboxylate ligand. MIL-53(Al) has a high thermal stability of up to 500 °C, which makes it a sought-after material in the MILs family [24]. MIL-53 (Al) exhibits a peculiar phenomenon known as "Breathing behavior" related to the hydrogen bonding interaction between the carboxylate groups of the BCD linker and the trapped water molecules, resulting in the contraction of its rhombic channels during

hydration or dehydration. Compared with the strong adsorption capacity of MIL-53(Al), MIL-53(Fe) has higher light absorption performance [25]. Nevertheless, its adsorption capacity is weak due to its small specific surface area. Even so, the unique "Breathing behavior" of MIL-53 (Figure 2c) has also brought more attention, which has also played a unique role in the field of multi-degradation [26].

### 2.2.4. MIL-88

MIL-88(Fe), a flexible iron-based material, exhibits excellent chemical and water stability. MIL-88A(Fe) (Figure 2d) comprises a divalent anion fumaric acid and a connected $Fe^{3+}$ octahedral trimer [27,28]. Despite the small surface area, the presence of unsaturated iron site would bring the activation ability towards $H_2O_2$ and strong adsorption and transport capacity. MIL-88B(Fe) has a three-dimensional net structure which is composed of Fe3-$\mu$3-oxo clusters connected by terephthalic acid [29]. It has good application potential and adaptability of photocatalysis [30]. MIL-88 has attracted much attention because of the excellent catalytic oxidation ability of nanomaterials derived from it due to the adjustable structural properties.

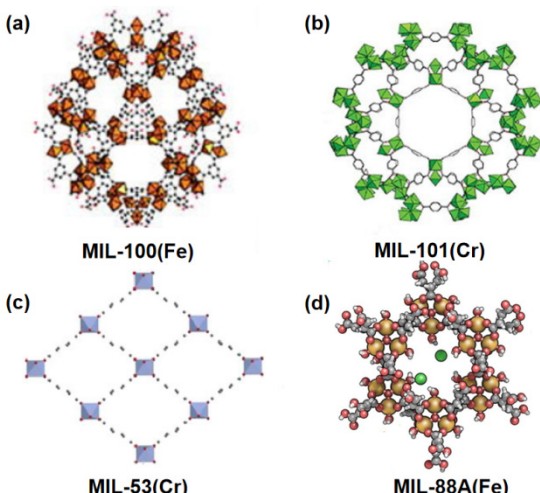

**Figure 2.** (**a**) The structure of MIL-100(Fe) [20]; (**b**) The structure of MIL-101(Cr) [22]; (**c**) The structure of MIL-53(Cr) [26]; and (**d**) The structure of MIL-88A(Fe) [28].

### 2.3. The Stability of MILs

Cyclic degradation experiments and material characterization can determine the stability of MILs before and after use. The stability of MILs usually determines its utilization value and practicability. Stability also becomes an important criterion to judge the quality of catalyst. Ren et al. conducted comparative experiments on MIL-53(Fe), MIL-100(Fe), and MIL-101(Fe) to test their stability and the relationship between stability and pH of degradation system [31]. The results showed that the three types of materials showed strong catalytic activity under strong acidic conditions but also resulted in poor material stability, which lost much catalytic activity after 20 cycles of experiments (Figure 3a–c). In addition, in the weak acidic and neutral conditions, although the catalytic activity of the three materials was not outstanding at the beginning, their catalytic activity decreased slowly with the progression of the cycle experiments. According to the characterization of the materials before and after recycling, new characteristic peaks appeared in MILs after several catalytic oxidation experiments, which the oxidation of the skeleton and the generation of new functional groups in the system may cause (Figure 3d–i). The stability of MILs before modification is not very good, especially if it is affected by pH in the system. Nevertheless, Qian et al. found that MIL-53(Al) maintained good stability in neutral and acidic solutions [32]. MIL-53(Al) showed good hydrolysis resistance and always retained pores in the degradation system.

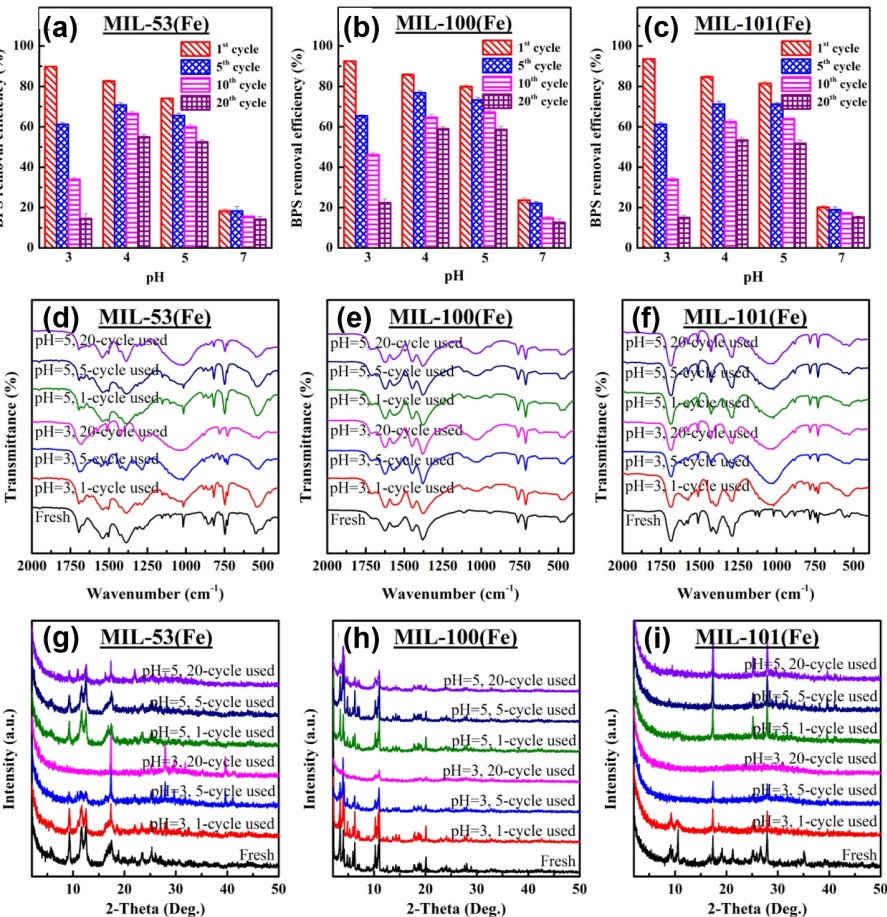

**Figure 3.** BPS removal efficiency in (**a**) MIL−53(Fe), (**b**) MIL−100(Fe), and (**c**) MIL−101(Fe) catalyzed Fenton-like processes at various pH levels; FTIR spectra of fresh and used (**d**) MIL−53(Fe), (**e**) MIL−100(Fe), and (**f**) MIL−100(Fe); XRD spectra of fresh and used (**g**) MIL−53(Fe), (**h**) MIL−100(Fe), and (**i**) MIL−101(Fe) [31].

## 3. Synthetic Techniques of MILs

MILs can be synthesized in various ways depending on time, temperature, and pH. Among them, hydrothermal/solvothermal is the most often used method because of its advantages, such as convenient operation. The synthesis methods of materials in most of the literature cited in this paper are the hydrothermal/solvothermal method. Of course, with the progress of research, more synthesis methods with different advantages have been explored. This section introduces the three commonly used synthesis methods and summarizes their advantages and disadvantages. In the selection process of the synthesis method, simple operation, low energy consumption, and good maintenance of crystallinity can be used as evaluation criteria. Undeniably, the practical application of the method is also worth considering.

### 3.1. Hydrothermal Synthesis

Hydrothermal synthesis is a common method for preparing MILs, usually in an autoclave (Teflon-lined) under constant conditions. Hydrothermal synthesis is superior to other synthesis technologies due to its characteristics of simple operation and high crystallization. For example, MIL-53(Cr) is usually synthesized by hydrothermal process. Eight grams of Cr(NO$_3$)$_3$•9H$_2$O and a certain amount of terephthalic acid and acetic acid were mixed and stirred 30 min with ultrasound [30]. Then, hydrofluoric acid was added, and the mixture was transferred to the hydrothermal reactor with the specific temperature to reaction. MIL-53(Cr) synthesized by hydrothermal synthesis has a high crystallinity. However, it takes a

long time and must use toxic fluorinated reagent. Large amounts of solvents are used in the synthesis process, and the specific surface area of MIL-53(Cr) prepared by hydrothermal method is small. Therefore, the product needs to be purified by post-treatment steps of high-temperature calcination. Therefore, the inherent defect of the hydrothermal synthesis method is the participation of toxic mineralizer (hydrofluoric acid) in the system. A small amount of hydrofluoric acid (HF) exists; investigating systems without HF has become a new direction to conquer. Without HF, MIL-100(Fe) was synthesized hydrothermally at room temperature [33]. First, after ultrasonic treatment for 15 min, $HNO_3$ and the iron precursor were dissolved in deionized water. Then, p-benzoquinone (promotes crystal growth) and 1,3,5-benzene tricarboxylic acid ($H_3BTC$) were added to the solution and stirred at room temperature for 12 h. After the MIL-100(Fe) was obtained, DMF, ethanol, NH4F, and hot deionized water were rinsed a few times and vacuum-dried at 423 K for 6 h. Removing toxic mineralizers is a positive guide for the synthesis of MILs by hydrothermal method. Nevertheless, there are still problems of insufficient crystallinity which need to be solved. The information of hydrothermal synthesis of some MILs is summarized in Table 1.

**Table 1.** Summary of information of MILs compounded by hydrothermal synthesis.

| Entry | MILs | Synthetic Material | Synthetic Condition | Synthetic Time | Ref. |
|---|---|---|---|---|---|
| 1 | MIL-53(Fe) | $FeCl_3 \bullet 6H_2O$ and $H_2BDC$ dissolved in DMF | 170 °C | 24 h | [34] |
| 2 | $NH_2$-MIL-101(Fe) | $FeCl_3 \bullet 4H_2O$ and $NH_2$–$H_2BDC$ | Room temperature | 24 h | [35] |
| 3 | MIL-101(Cr) | $Cr(NO3)_3 \bullet 9H_2O$, HF and $H_2BDC$ | 493 K | 8 h | [36] |
| 4 | MIL-88A(Fe) | $FeCl_3 \bullet 6H_2O$ and $C_4H_4O_4$ dissolved in anhydrous ethanol | Room temperature | 24 h | [37] |
| 5 | $SO_3H$-MIL-101(Cr) | 1,3-propanesultone and imidazole dissolved in ethanol solvent | 50 °C | 24 h | [38] |
| 6 | MIL-88(Fe, Ni) | $FeCl_3 \cdot 6H_2O$, $Ni(NO_3)_2 \cdot 6H_2O$ and TPA dissolved in DMF and NaOH | 100 °C | 48 h | [39] |
| 7 | MIL-100(Cr, Fe) | MIL-100 (Cr): chromium (III), trimesic acid and DI water MIL-100(Fe): $FeCl_3 \cdot 6H_2O$, trimesic acid and DMF | 493 K and 423 K | 15 h and 20 h | [40] |
| 8 | OH-MIL-53(Al) | $AlCl_3 \bullet 6H_2O$ and BDC-OH in DMF | 125 °C | 8 h | [41] |
| 9 | MIL-100(Fe) | $FeSO_4 \bullet 7H_2O$ and $H_3BDC$ with NaOH | Room temperature | 24 h | [42] |
| 10 | $NH_2$-MIL-53(Al) | $AlCl_3 \bullet 6H_2O$ and $NH_2$-BDC dissolved in DMF | 130 °C | 72 h | [43] |

### 3.2. Steam-Assisted Conversion

The steam-assisted method utilizes heat and mass transfer during the reaction to synthesize materials. Because it does not contain toxic mineralizers and guarantee the material's crystallinity, importance has been attached to it for its utility to researchers. It is a very suitable method for large-scale industrial synthesis. The experiment is usually carried out at room temperature by placing raw materials in water or organic solvent, allowing them to age naturally. Consequently, the organic components and inorganic components can react with each other and assemble into metal–organic frame materials with high crystallinity. For example, MIL-100(Cr) can be synthesized by steam-assisted conversion. $CrCl_3 \bullet 6H_2O$ (2 mM) and $H_3BTC$ (1 mM) were mixed and easily ground at room temperature; a sieve plate provided with vents was supported by a holder [44]. They were then placed in a 100 mL Teflon-lined autoclave while 10 mL deionized water was carefully fed into the bottom of the container to ensure that steam combined with the precursor at 423 K for 9 h. MIL-100(Cr) was finally produced and then centrifuged 3–4 times with $H_2O$. The experimental device is shown in Figure 4. The result showed that the MIL-100(Cr) synthesized had no superfluous ligands existing. Therefore, in order to obtain a high amount of production, further research must be conducted on a broader and deeper level to optimize the utilization of this technique.

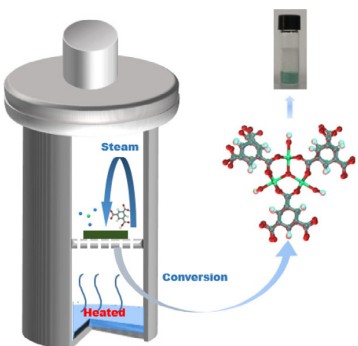

**Figure 4.** A schematic diagram of steam-assisted conversion of MIL-100(Cr) [44].

### 3.3. Microwave-Assisted Synthesis

Microwave synthesis (MW) is a new technology for synthesizing nanomaterials. It can be used at the molecular level to increase yields on the basis of the stimulation of rapidly heating materials, so it is emerging in MILs synthesis research. Dong et al. prepared MIL-53(Fe) by a microwave-assisted approach [45]. A mixed solution of $FeCl_3 \bullet 6H_2O$ (0.45 g), $H_2BDC$ (0.137 g) and DMF (10 mL) was transferred to a Teflon tube. Then, the Teflon tube was placed in a microwave oven and heated to 150 °C for 15 min. The resulting orange suspension was purified by centrifugation of dimethyl formamide (DMF) and ethanol at 60 °C for 1 h. The material was then dried at 50 °C in a vacuum to obtain an orange powder with 30% yield. It is worth emphasizing that MIL-53(Fe) synthesized in this paper had better phase selectivity than that synthesized by the conventional electric (CE)-heating-based solvothermal approach. This may depend on the smaller crystallinity of the material and the pure octahedral structure resulting from the MW method. However, due to its high energy consumption and expensive equipment, coupled with less research, its practical application is limited.

### 4. Modification and Activation of MILs

MILs have become substitutes for traditional heterogeneous catalysts due to their unique structural characteristics, suitable stability, and relatively simple synthesis. However, in the study of the degradation of organic pollutants, MILs have some deficiencies, such as the absence of active sites and poor thermal stability, so it needs modification. We summarize the common methods of MILs modification herein in recent years: (1) metal sites, (2) ligands modification, and (3) activation. Table 2 lists the degrading information of three methods modified MILs in the Fenton system.

**Table 2.** Summary of modified MILs and their degrading information in the Fenton system.

| Entry | MILs | Characterization Result | Target Compound | Removal Efficiency | Photo-Fenton Conditions | Ref. |
|---|---|---|---|---|---|---|
| 1 | MIL-101(Fe$^{II}_3$, Mn) |  | Phenol | >99% (10 min) | catalyst, 100 mg/L; pH = 6; $H_2O_2$, 20 mM; Temp. 35 °C | [46] |
| 2 | Fe$_{0.75}$Cu$_{0.25}$(BDC) |  | SMX | 100% (120 min) | catalyst, 500 mg/L; pH = 5.6; $H_2O_2$, 6 mM; Temp. 25 °C | [47] |
| 3 | L-MIL-53(Fe, Mn) |  | CIP | >89% (30 min) | catalyst, 100 mg/L; pH = 7; $H_2O_2$, 5 mM; Temp. 25 °C; visible light | [48] |

**Table 2.** *Cont.*

| Entry | MILs | Characterization Result | Target Compound | Removal Efficiency | Photo-Fenton Conditions | Ref. |
|-------|------|------------------------|-----------------|--------------------|-----------------------| -----|
| 4 | MIL-53(Fe, Ni) |  | RhB | >93% (180 min) | catalyst, 300 mg/L; pH 4~12; PDS, 0.1 mM; Temp. room temperature | [49] |
| 5 | nZVI/MIL-101(Cr) |  | TC | >90% (120 min) | catalyst, 250 mg/L; pH = 7; $H_2O_2$, 50 mM; Temp. room temperature | [21] |
| 6 | CUCs-MIL-88B(Fe)/$Ti_3C_2$ |  | SMX | >90% (120 min) | catalyst, 200 mg/L; pH = 3; $H_2O_2$, 10 mM; Temp. room temperature; visible light | [50] |
| 7 | $NH_2$-MIL-101(Fe) |  | BPF | >90% (40 min) | catalyst, 300 mg/L; pH = 6.2; PS, 1 mM; Temp. 25 °C | [51] |
| 8 | $NH_2$-MIL-101(Fe) |  | BPA | >95% (30 min) | catalyst, 200 mg/L; pH = 6; $H_2O_2$, 10 mM; Temp. 30 °C | [35] |
| 9 | $Bi_2WO_6$/ $NH_2$-MIL-88B(Fe) |  | TC | >89% (130 min) | catalyst, 350 mg/L; pH = 4; Temp. room temperature; visible light | [52] |
| 10 | $MoS_2$/ $NH_2$-MIL-88B(Fe) |  | TC | >96%(30 min) | catalyst, 500 mg/L; PH = 7; Temp. 35 °C; visible light | [53] |
| 11 | CUS-MIL-100(Fe) |  | SMT | 100% (180 min) | catalyst, 500 mg/L; pH = 4; $H_2O_2$, 6 mM; Temp. 25 °C | [54] |
| 12 | CUS-MIL-250 |  | TC-HCL | >95% (80 min) | catalyst, 200 mg/L; pH = 4; $H_2O_2$, 2 mM; Temp. room temperature; visible light | [55] |
| 13 | CUMSs/ MIL-101(Fe, Cu) |  | CIP | 100% (30 min) | catalyst, 100 mg/L; pH = 7; $H_2O_2$, 3 mM; Temp. 25 °C | [56] |

### 4.1. Metal Sites

4.1.1. Bimetal Sites

In the process of MILs development, bimetallic MIL materials have become a common modification method. Bimetallic MIL materials can not only improve the catalytic shortcomings caused by one metal but also produce synergistic effects to promote the catalytic degradation rate. Fe-MILs usually exhibit low catalytic performance and stability due to their low porosity and easy aggregation at high reaction temperature. Therefore,

researchers often add another metal to improve the material and compensate for shortcomings. Our research group prepared MIL-101(Fe$^{II}_3$, Mn) by adding Mn to MIL-101(Fe), which can be used to activate H$_2$O$_2$ as another reaction center to degrade phenol (Figure 5a) [46]. Moreover, the electrons around Mn were transferred mainly to the Fe atomic region because of the difference in electronegativity, so H$_2$O$_2$ was absorbed mainly in the electron-rich region, which finally improved the utilization rate of H$_2$O$_2$ (81.2%). Ma et al. synthesized a series of novel Fe-Mn mixed oxide hollow microspheres using a hard template method and applied to catalytic oxidation of 1,2-dichlorobenzene (DCB) [57]. The results indicated that catalytic performance of Fe-Mn mixed oxide hollow microspheres was much higher than that of Fe-MIL MOF. Furthermore, the optimal FeMn20 hollow microspheres (Mn/(Fe + Mn) $\approx$ 20%) exhibited the highest catalytic performance at a low temperature of 400 °C. More importantly, this catalyst exhibited good Cl-resistant ability, strong water-resistant ability, and excellent catalytic stability. It was a very successful bimetallic material with a large BET specific surface area and high surface-active oxygen concentration. Tang et al. prepared an iron and copper bimetallic MOF material (Fe$_X$Cu$_{1-X}$(BDC)) by a simple solvothermal method for degrading sulfamethoxazole (SMX) in a Fenton reaction system (Figure 5b) [47]. Fe and Cu species were active sites for H$_2$O$_2$ activations and had a synergistic effect on the generation of •OH radicals, which directly led to the degradation of SMX. Therefore, the bimetallic modification resulted in better degradation performance of the Fe$_X$Cu$_{1-X}$(BDC)/H$_2$O$_2$ system. Furthermore, Fe$_{0.75}$Cu$_{0.25}$(BDC) had satisfactory reusability and could be reused many times. Wu et al. prepared iron–nickel bimetallic organic frameworks (FeNi$_X$-BDC, H$_2$BDC: terephthalic acid), which could degrade organic dyes, such as MB and MO (Figure 5c) [58]. The addition of Ni increased the specific surface area of the material and further reduced the surface charge of the material. This Fe-Ni bimetallic material exhibited excellent degradation performance for organic dyes and good repeatability. Notably, this system was suitable for a wider pH working range (3–9). Šuligoj et al. discussed a novel bimetal Cu-Mn porous silica-supported catalyst to decompose methylene blue (Figure 5d) [59]. The addition of copper extremely reduced the leaching of Mn from the porous silica carrier and improved the degradation efficiency as an additional surface adsorption site.

### 4.1.2. Metal Nanoparticle Doping

Many nano-metals have strong activity, and the composite materials will have a better catalytic degradation effect by combining them with organic frameworks. For instance, nano zero-valent iron (nZVI) is a type of particle with high activity and easy aggregation. It needs support from porous materials to improve its stability. Hou et al. modified MIL-101(Cr) with nZVI through impregnating nZVI to MIL-101(Cr) for adsorption and degradation of tetracycline (TC) (Figure 6a,b) [21]. MIL-101(Cr) provided a huge surface area for nZVI coating, thus increasing the adsorption capacity of TC, to a certain extent. Fortunately, nZVI/MIL-101(Cr) also had similar Fenton-like reactivity and less iron leaching. In addition, the combination of framework and nano-metal could also effectively increase the Lewis acid site of the material. The result showed that nZVI/MIL-101(Cr) degraded 90% of the dye within 120 min. Ahmad et al. successfully synthesized MIL-88B(Fe) with mixed-valence coordinatively unsaturated iron centers on ultrathin Ti$_3$C$_2$ nanosheet for removal of contaminants (Figure 6c,d) [50]. The formation of composite material improved the efficiency of electron transmission and brought excellent degradation performance in the visible light. The addition of bimetallic modification can also effectively solve the problem of slow conversion between Fe(II) and Fe(III) of activated MILs in the reaction system.

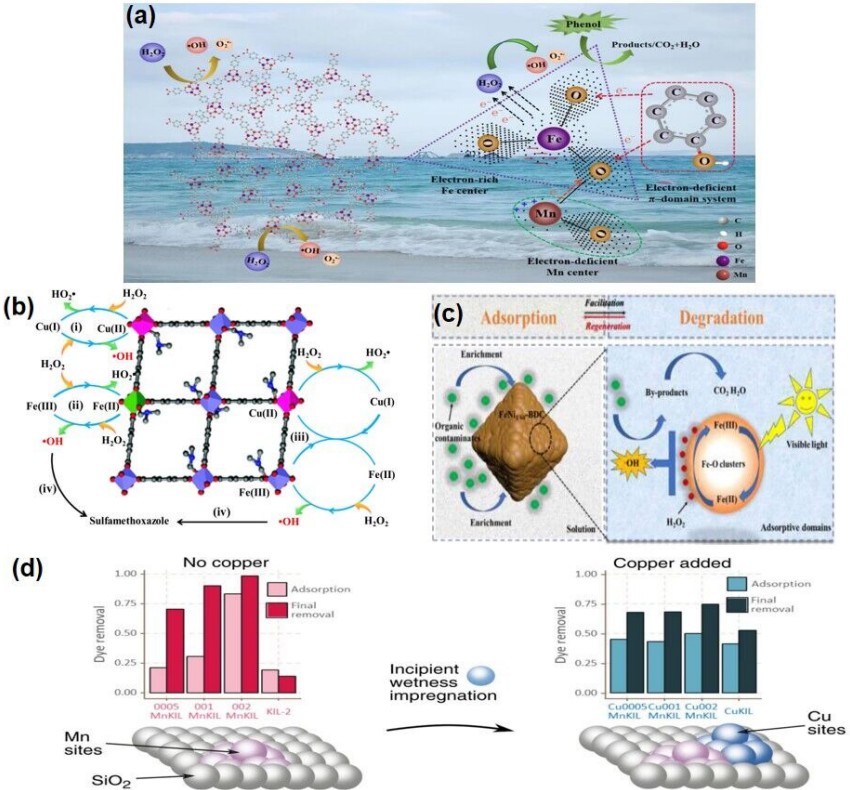

**Figure 5.** (**a**) Schematic for the reaction mechanism in MIL-101($Fe^{II}_3$, Mn)/$H_2O_2$/phenol system [46]; (**b**) Schematic illustration of SMX removal in the $Fe_{0.75}Cu_{0.25}$(BDC)/$H_2O_2$ system [47]; (**c**) Reaction mechanism of contaminant degradation in $FeNi_{1/60}$-BDC/$H_2O_2$/visible light system [58]; and (**d**) Schematic illustration of CuMnMIL and comparison before and after degradation [59].

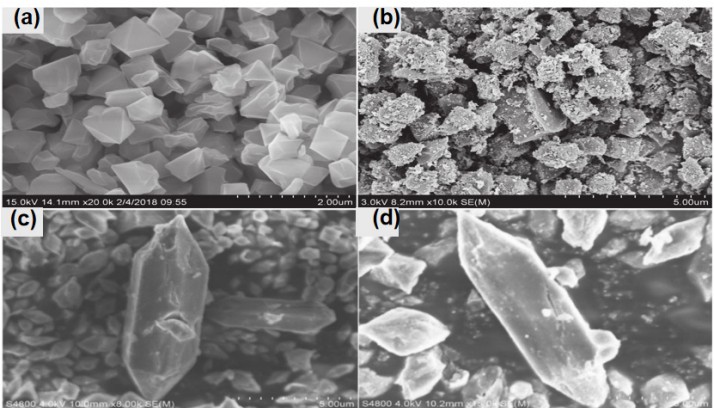

**Figure 6.** The SEM images of (**a**) MIL-101(Cr) [21], (**b**) nZVI/MIL-101(Cr) [21], (**c**) MIL-88B-Fe, and (**d**) CUCs-MIL-88B-Fe [50].

### 4.2. Ligand

The introduction of ligands, such as $-NH_2$, $-SO_3H$, $-CH_3$, and -F, can improve the degradation efficiency of the parent materials without changing the topological structure. The addition of ligands can regulate the surface properties of the materials and improve adsorption capacity. Fluorinated MIL-101(Cr) proved to have too high thermal stability and increased porosity, which greatly improved the adsorption capacity of benzene [60]. In the last five years of research, amino-functionalized materials have been the most common materials to enhance the degradation efficiency of organic pollutants by MILs. Introducing $-NH_2$ can increase the electron density and accelerate the electron transfer velocity in the reaction. Thus, an •OH radical was continuously produced to degrade organic pollutants.

Liu et al. used NH$_2$-MIL-101(Fe) in the persulfate activated system to degrade bisphenol F (BPF) (Figure 7a) [51]. The degradation efficiency was greater than parent MIL-101 (Fe), and it had a wider pH adaptation range and better anti-anion interference ability. Moreover, we synthesized NH$_2$-MIL-101(Fe) at room temperature for the degradation of bisphenol A (BPA) (Figure 7b) [35]. By regulating the Fe-oxo node, -NH$_2$ caused Fe(II) to form the in situ distribution in the MIL. The result showed that the degradation rate of BPA by NH$_2$-MIL-101(Fe) was also much higher than that by MIL-101(Fe). It can be seen that amino-functionalized MILs can truly bring better and remarkable results in degrading organic pollutants.

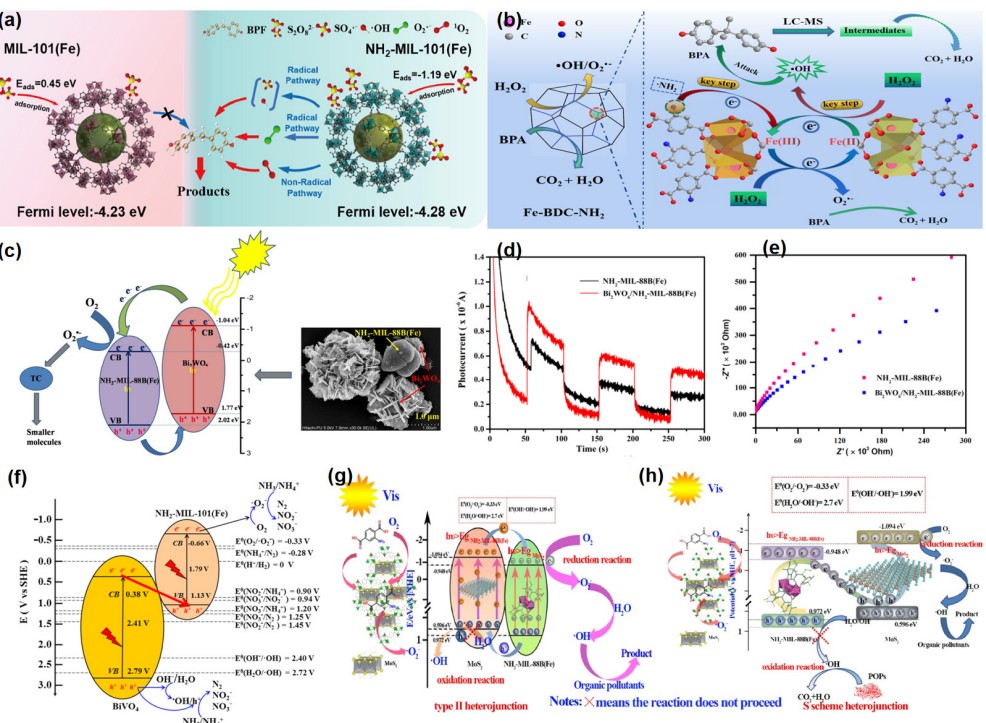

**Figure 7.** (**a**) Schematic illustration of mechanism of BPF removal in NH$_2$-MIL-101(Fe)/PS or MIL-101(Fe)/PS systems [51]; (**b**) Schematic illustration of mechanism of BPA removal in Fe-BDC-NH$_2$/H$_2$O$_2$ system [35]; (**c**) Proposed photocatalytic mechanism of Bi$_2$WO$_6$/NH$_2$-MIL-88B(Fe) composite for the degradation of TC [52]; (**d**) Transient photocurrent response [52]; (**e**) EIS spectra of NH$_2$-MIL-88B(Fe) and Bi$_2$WO$_6$/NH$_2$-MIL-88B(Fe) [52]; (**f**) Mechanism for ammonia removal [61]; (**g**) type II photocatalytic mechanism for NH$_2$-MIL-88B(Fe)/MoS$_2$ nanocomposite under Xenon lamp irradiation [53]; and (**h**) S-scheme photocatalytic mechanism for NH$_2$-MIL-88B(Fe)/MoS$_2$ nanocomposite under Xenon lamp irradiation [53].

Amino-functionalized MILs have also been used to synthesize heterogeneous structures with inorganic materials in many studies. The formation of heterostructures has been shown to facilitate electron transfer in catalysis, and more importantly, it enhances the light capture capacity of photogenerated carriers. So, this type of material was widely used in the field of photocatalysis. Kaur et al. successfully synthesized a novel Bi$_2$WO$_6$/NH$_2$-MIL-88B(Fe) heterostructure for the decomposition of tetracycline (TC) [52]. With the help of visible light, the TC degradation efficiency of Bi$_2$WO$_6$/NH$_2$-MIL-88B(Fe) was 89.4% higher than that of the parent NH$_2$-MIL-88B(Fe); the materials showed a longer carrier lifetime, and there was higher charge separation in the system (Figure 7c–e). Shi et al. fabricated a Z-scheme NH$_2$-MIL-101(Fe)/BiVO$_4$ heterostructure to degrade ammonia nitrogen in the photocatalytic system as well (Figure 7f) [61]. Compared with other heterostructures, the Z-scheme heterostructure brought the photogenerated carriers stronger redox ability and enhanced the degradation efficiency of ammonia nitrogen. Feng et al. also took advantage

of the fact that heterogeneous structures in the reaction system could not only enhance the various properties of photogenerated carriers but also improve the non-surface area of the material (Figure 7g,h) [53]. They prepared $NH_2$-MIL-88B(Fe)/$MoS_2$ for degrading tetracycline hydrochloride (TC) availability. They prepared $NH_2$-MIL-88B(Fe)/$MoS_2$ to evaluate the activity in degrading tetracycline hydrochloride (TC).

In the study on ligand modification of MILs, few other ligands appeared, and only a few ligands showed excellent degradation properties when combined with parent materials. Gao et al. synthesized MIL-88B(Fe)-X (X=$NH_2$, $CH_3$, H, Br, and $NO_2$) and used the addition of ligands to reduce the electron density in the reaction for solving the problem of slow Fe(III) reduction in the Fenton system (Figure 8a) [62]. The results showed that MIL-88B(Fe)-$NO_2$, with the substituent of highest electrophilicity among these catalysts, had an excellent performance in the degradation of phenol. In addition, -$SO_3$H can also make a contribution to the production of Brønsted acid sites and Lewis acid sites for enhancing the degradation efficiency. $SO_3$H-MIL-101(Cr) was prepared for the conversion of oleic acid (94.3%) with the synergistic effect between Brønsted acid and Lewis acid. Mortazavi et al. used MIL-101(Cr) to construct MIL-pip-$SO_3$H and MIL-DABCO-$SO_3$H (Figure 8b,c) [63]. As a Brønsted solid acid catalyst, MIL-pip-$SO_3$H and MIL-DABCO-$SO_3$H were good at degrading styrene oxide and better than any Lewis acid catalysts that have been reported. In general, there are few studies on modifying MILs in the Fenton system by ligands other than -$NH_2$, which we can work on in the future.

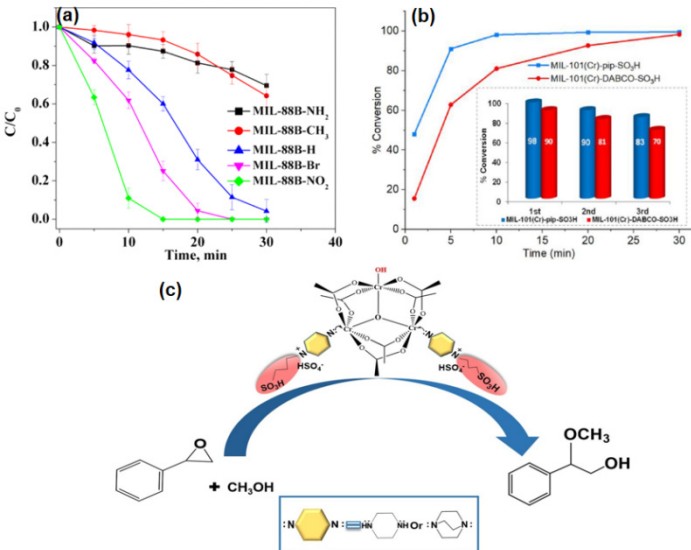

**Figure 8.** (**a**) Time profiles of phenol degradation during MIL-88B(Fe)-X catalysis. Fenton-like reaction [62]; (**b**) Kinetic profiles of the methanolysis of styrene oxide [63]; and (**c**) Mechanism for MIL-pip-$SO_3$H and MIL-DABCO-$SO_3$H [63].

### 4.3. Activation

In recent years, increasingly more studies have shown that controlled decomposition of MILs (e.g., pyrolysis) can effectively improve its catalytic activity. It gives MILs a new mission to transform into more stable materials that conventional processes cannot synthesize. The treated MILs retained the traditional advantages of high stability and high metal load and obtained more reaction sites due to the unsaturated ligands brought by pyrolysis. Tang et al. synthesized MIL-100(Fe) with Fe(II)/Fe(III) mixed-valence co-ordinatively unsaturated iron center (CUS-MIL-100(Fe)) for enhancing degradation of sulfamethazine (Figure 9a,b) [54]. The incorporation of Fe(II) and Fe(III) CUSs resulted in large specific surface area, as well as the formation of mesopores, which helped the CUS-MIL-100(Fe) to exhibit a higher removal efficiency of SMT than MIL-100(Fe). Guo et al. prepared MIL-100(Fe) mixed CUSs with different $Fe^{2+}$/$Fe^{3+}$ ratios by heat treatment for degrading tetracycline hydrochloride (TC-HCL) (Figure 9c,d) [55]. The formation of

CUSs expanded the pore volume so that the material could better absorb visible light and produce more active oxidizing substances to improve the degradation performance. In addition, a good synergistic effect between CUCs and $H_2O_2$ in the reaction system had been observed, ascribed to enhanced adsorption and activation of $H_2O_2$ on the CUCs Lewis acid sites. Moreover, Liang et al. prepared CUMSs/MIL-101(Fe, Cu) for the degradation of ciprofloxacin (CIP) (Figure 9e) [56]. CUMSs/MIL-101(Fe, Cu) inherited the synergistic effect of Fe-Cu bimetallic MILs combined with the construction of mixed valence of Fe(II)/Fe(III) and Cu(I)/Cu(II) as coordinatively unsaturated metal sites (CUMSs), and good catalytic oxidation performance was demonstrated.

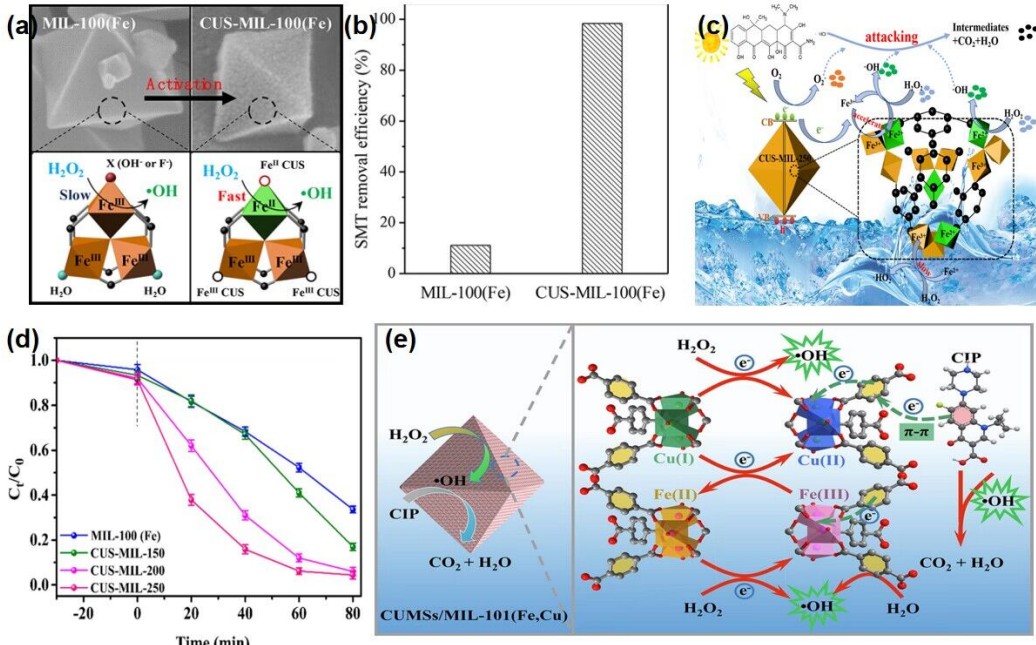

**Figure 9.** (**a**) Schematic illustration of the $H_2O_2$ activation by CUS−MIL−100(Fe) [54]; (**b**) The SMT removal efficiency by MIL−100(Fe) and CUS−MIL−100(Fe) [54]; (**c**) Schematic illustration of reaction mechanism in CUS−MIL−250/$H_2O_2$/Vis reaction system [55]; (**d**) Degradation of TC–HCl [55]; and (**e**) Mechanism for CIP removal [56].

## 5. The Applications of MILs in Fenton Reaction System

In 1894, the French chemist Fenton found that $Fe^{2+}$ and $H_2O_2$ can effectively oxidize tartric acid in an acidic environment [64]. Later, in 1964, Eisenhouser used Fenton reagent ($Fe^{2+}$ and $H_2O_2$) to treat phenol and alkyl benzene wastewater, setting a precedent for the application of Fenton reagent in the treatment of environmental pollution. The emergence and development of iron-based MOFs have also expanded the application of Fenton reaction system to degrade organic pollutants [65]. MILs have become the ideal catalytic materials for the Fenton reaction system to degrade organic pollutants because of their unique structural characteristics, such as appropriate stability, large specific surface area, and adjustable pore structure [66]. MILs have been frequently utilized in catalysis as artificial enzymes due to their high density of transition metal sites, large specific surface area, and high stability. We found that MIL-101(FeII) nanozyme exhibited peroxidase-like activity, and it could be used to catalyze $H_2O_2$ for oxidation of N,N-Diethyl-p-phenylenediamine sulfate salt (DPD) (Figure 10) [67]. We used $Fe^{2+}$ to control the formation of (FeII)-oxo nodes so that the material showed better catalytic activity. At the same time, we proposed a MIL-101(FeII) colorimetric biosensor and used it to detect $H_2O_2$ and glucose. This work opens up an opportunity for MIL-101(FeII) as a promising biosensor in bioanalysis, food safety, and environmental monitoring. With the deepening of research on the organic contaminant removal by the Fenton reaction system, MILs have also been applied in electro-Fenton and photo-Fenton reaction systems.

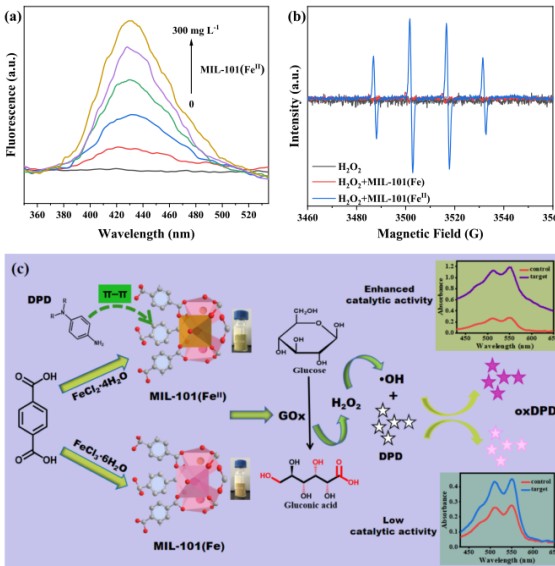

**Figure 10.** (**a**) Fluorescent measurements, (**b**) EPR measurement of •OH, and (**c**) Colorimetric determination of $H_2O_2$ and glucose based on MIL−101(FeII) catalysis [67].

## 5.1. Application of MILs in Electro-Fenton Reaction System

The electro-Fenton reaction system generates $Fe^{2+}$ or $H_2O_2$ continuously through electrochemical action, which can generate •OH to degrade organic pollutants. Because the electro-Fenton anode can efficiently produce free and highly active •OH, the cathode uses electrons to react with redox to produce $H_2O_2$, which combined with iron-based MOF, forms a complete electro-Fenton system. In the electro-Fenton system, the redox cycle of $Fe^{2+}/Fe^{3+}$ is as shown in Equations (1) and (2). Moreover, water molecules are oxidized to •OH at the anode, while $O_2$ is reduced to $H_2O_2$ at the cathode (Equations (3) and (4)). Electron transfer also promotes the reduction of $Fe^{3+}$ to $Fe^{2+}$ (Equation (5)). The reaction system has the merits of strong oxidation capacity, low power consumption, and good environmental compatibility [68].

$$H_2O \rightarrow \bullet OH + H^+ + e^- \tag{3}$$

$$O_2 + H^+ + 2e^- \rightarrow H_2O_2 \tag{4}$$

$$Fe^{3+} + e^- \rightarrow Fe^{2+} \tag{5}$$

However, due to the leaching and decrease of active sites, the regeneration of unsaturated sites slows down with the progress of the electro-Fenton reaction. Therefore, it is usually considered to use a high-pore catalyst for improvement. Priyadarshini et al. successfully synthesized MIL-53(Fe)@Fe$_3$O$_4$@C for degradation of salicylic acid in water in an electro-Fenton system [69]. Ye et al. synthesized MIL-88B(Fe)/Fe$_3$S$_4$ hybrids via a facile sulfurization to treatment trimethoprim (TMP) in the heterogeneous electro-Fenton (HEF) system (Figure 11a) [70]. The hybrid molecular orbitals of S-Fe improved the absence of coordinated unsaturated sites in the electro-Fenton system of conventional MIL-88B(Fe), which brought a larger expansion room to release the overlapping orbital parts between iron sites and $H_2O_2$. Consequently, it promoted the reaction's electron transfer from $H_2O_2$ to Fe(III). The result showed that the reaction system removed TMP quickly at mild pH. It also confirmed that MIL-88B(Fe) sulfidation could enhance the catalyst activity. In addition, the absence of catalyst active sites and weak electron transport in the electro-Fenton system could be solved by metal–carbon hybrids and the introduction of the other metal. Du et al. prepared MIL-53(Fe)@MoO$_3$ to degrade sulfamethazine (SMT) and studied the corresponding pyrolysis catalysts involving FeMo-based bimetallic and porous carbon (PC) (FeMo@PC) (Figure 11b) [71]. The addition of PC enhanced the electron transfer in the

system and accelerated the synergistic effects among Fe, Mo, and carbon, which caused the FeMo@PC-2 ($MoO_3$ with 30.0 mg) to exhibit excellent performance for SMT removal.

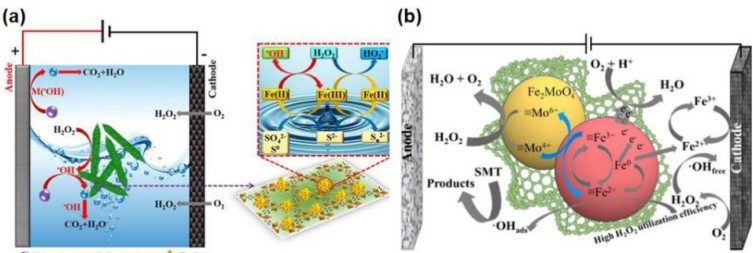

**Figure 11.** (**a**) Schematic illustration of the reaction mechanism for the S−MIL−88B−3h−3−catalyzed degradation of TMP [70]; (**b**) Schematic illustration of the mechanism in the FeMo@PC−2/Hetero-EF process [71].

### 5.2. Application of MILs in Photo-Fenton Reaction System

The photo-Fenton reaction system uses ultraviolet wavelength radiation to enhance the oxidation ability of the Fenton reagent, thus accelerating the degradation of organic pollutants [72,73]. The advantages of photocatalysis and Fenton-like technology grant MILs a wide range of applications in the field of photo-Fenton. The photo-Fenton system is considered one of the most promising processing technologies because it is easy to operate without high temperature and pressure [74]. Zhang et al. designed a novel catalyst a-$Fe_2O_3$/MIL-53(Fe), which had the advantages of thermodynamic stability of a-$Fe_2O_3$, appropriate bandgap width, and the synergism of atom/MOFs catalyst (Figure 12a) [75]. At the same time, forming a heterojunction structure reduces the band gap, strengthens visible light's absorption ability, and improves photocarrier transmission. a-$Fe_2O_3$/MIL-53(Fe) hydrochloride showed excellent degradation performance of Tetracycline hydrochloride (TC-HCl) in the photo-Fenton system. Li et al. used CA to modify MIL-88A(Fe) for degrading methylene blue (MB) and carbamazepine (CBZ) (Figure 12b) [76]. The modification of CA could accelerate the separation and transfer of photo-charge and optimize the Fe(II)/Fe(III) cycle, thus improving the catalytic performance. The photocatalytic rate of the improved material is up to 6.5 and 2.5 times higher than that of the raw material. Moreover, the catalyst shows excellent stability and repeatability. We have compared many photo-Fenton systems by MILs in Table 3 with their target compound, removal efficiency and conditions as well as the $S_{BET}$ area of MILs.

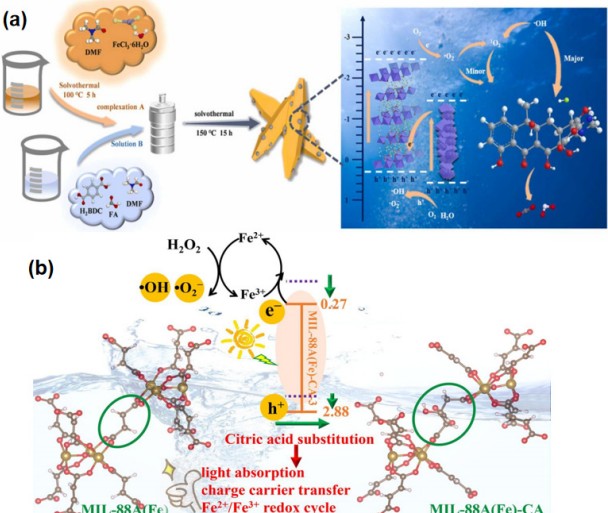

**Figure 12.** (**a**) Schematic illustration of the reaction mechanism for the TC-HCl removal [75]; (**b**) Schematic illustration of the mechanism in the MIL-88a(Fe)-CA/photo-Fenton process [76].

**Table 3.** Summary of MILs in photo-Fenton reaction.

| Entry | MILs | $S_{BET}$ Area $m^2/g$ | Target Compound | Removal Efficiency | Photo-Fenton Conditions | Ref. |
|---|---|---|---|---|---|---|
| 1 | MIL-88A(Fe)-CA | - | MB and CBZ | >97% | catalyst, 125 mg/L; pH = 3; $H_2O_2$, 25 mM; Temp. 40 °C | [76] |
| 2 | MIL-53(Fe) | - | SMX | >96% (120 min) | catalyst, 2 mmol/L; pH = 4; $H_2O_2$, 2 mM; Temp. room temperature | [77] |
| 3 | N-BiFeO$_3$/ NH$_2$-MIL-53(Fe) | - | TCH | >99% (60 min) | catalyst, 200 mg/L; pH = 7; $H_2O_2$, 0.2 mL/L; Temp. room temperature | [78] |
| 4 | Zn/Co-ZiFs@ MIL-101(Fe) | 376 | RhB | >95% (3 h) | catalyst, 200 mg/L; pH = 5; $H_2O_2$ 90 mM; Temp. 5 °C | [79] |
| 5 | Cu$_2$O/ MIL(Fe/Cu) | 1553 | TCL | >80% (80 min) | catalyst, 50 mg/L; pH = 7; $H_2O_2$, 49 mM; Temp. 25 °C | [80] |
| 6 | Bi$_{3.64}$Mo$_{0.63}$O$_{6.55}$/ MIL-88A(Fe) | 80.196 | TCH | >84% (30 min) | catalyst, 500 mg/L; pH = 7; $H_2O_2$ 2.5 mL/L; Temp. room temperature | [81] |
| 7 | MIL-88A(Fe) | 13.17 | OFL | 100% (50 min) | catalyst, 250 mg/L; pH = 7; $H_2O_2$ 1 mL/L; Temp. room temperature | [82] |
| 8 | CuS/MIL-Fe | 914.19 | APAP | 100% (30 min) | catalyst, 200 mg/L; pH = 5; $H_2O_2$ 15 mM; Temp. room temperature | [83] |
| 9 | MIL-53(Fe)/BiOI | - | TC | >86% (14 min) | catalyst, 200 mg/L; pH = 7; $H_2O_2$ 10 mM; Temp. room temperature | [84] |
| 10 | Fe$_3$S$_4$ | - | SMX | 100% (10 min) | catalyst, 300 mg/L; pH = 5; $H_2O_2$ 0.2 mL/L; Temp. room temperature | [85] |
| 11 | NH$_2$ -MIL-88B(Fe) | - | ACTM | 100% (40 min) | catalyst, 140 mg/L; pH neutral; $H_2O_2$ 0.21 mL/L; Temp. 25 °C | [86] |

## 6. Summary and Outlook

Due to their large specific surface area, high red crystallinity, and controllable ordered pore structure, MILs have become a hot spot in the research field of MOF materials [22]. In this review, we systematically introduced the development of MILs in Fenton reaction to degrade organic pollutants, the classification of MILs which have been applied in Fenton catalytic systems, the common synthesis methods of MILs and their advantages and disadvantages, the modification and activation of MILs, and the application of MILs in Fenton, electro-Fenton, and photo-Fenton reaction.

At present, researchers are trying to improve MILs in more ways to optimize their poor thermal and mechanical stability and to enhance their degradation efficiency while maintaining their original structural properties. Researchers are investigating combining MIL materials with other materials to obtain composite materials with excellent performance. However, there are also some problems in the study of MILs:

(1) Few studies have conducted long-term stable tests on the developed materials and the reaction system, so we do not know whether many MIL materials have long-term stability and circularity.

(2) Many MILs face the problem of inactivation after repeated use and need to be regenerated. However, more suitable regeneration methods need to be further studied.

(3) Due to the suitable chemical and mechanical stability of some MILs, as well as the limitations of many synthesis methods, some MILs are difficult to be used in real life and mass production.

On the basis of the above problems, we should continue to explore and strive to find suitable catalytic applications for MILs. Of course, it can be predicted that MILs will have attractive application prospects in the development of new functional materials, such as gas storage, heterogeneous catalysis, drug storage and release, optical, electrical, magnetic, and so on.

**Author Contributions:** Conceptualization, Q.C.; formal analysis, K.X., B.S., Q.C., P.H., K.L., L.W., S.W. and L.C.; writing—original draft preparation, K.X.; writing—review and editing, B.S., Q.C., K.L. and L.C.; supervision, project administration, and funding acquisition, Q.C. All authors have read and agreed to the published version of the manuscript.

**Funding:** This project was financially supported by the National Natural Science Foundation of China (Grant No. 51672312); and Fundamental Research Funds for the Central Universities, South-Central Minzu University (Grant Nos. CZQ23007 and KTZ20043).

**Data Availability Statement:** Not applicable.

**Conflicts of Interest:** There are no conflict of interest to declare.

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
