# Peer review of "Recent Progress of MIL MOF Materials in Degradation of Organic Pollutants by Fenton Reaction"

_catalysts, doi:10.3390/catal13040734_

Round 1

Reviewer 1 Report

Xiao K. et al., analyzes and summarizes the progress of MILs in degradation of organic pollutant by Fenton reaction in recent years. MIL-MOFs materials are very promising in the field of degradation of organic pollutants, and a review with the recent progresses is quite an interested. The review it seems well-organized but the authors must correct some points and add more information,  in order to be improved the quality of the text. More specifically;

1) Some English expressions should be removed, like ''improvement of living standards'', ''the researchers found that'', ''Fenton reaction is as reaction 1''. Generally the authors should scrutinize whole the text for the English language and write the text to a more formal (scientific) way

2) It should be better, if the authors add more Figures, i.e in the synthetic techniques. Also the addition of one or two tables could help the readers to take more information about the field

3) The introduction must be written more extensively and the authors include more information. I think that it is too shorten for review manuscript

4) Paragraph 4.2: Ligand. Are there more ligands except from NH2? the authors should mention, and if not why? 

5) Paragraph 5: Applications. The authors should mention more information about electro-fenton and photo-fenton, because this is the 'core' of  the review.

6) It would be better if it is not mentioned so many times, the expression.. ''our research group'', one time is enough

Author Response

Responses to Reviewers’ Comments

Dear Editors ad Reviewers:

Thank you for your letter and for the reviewer’s comments concerning our manuscript entitled “Recent process of MIL MOF materials in degradation of organic pollutants by Fenton reaction”. The authors gratefully acknowledge the reviewers for your helpful comments. We have carefully revised our manuscript according to your comments and remarks. Our responses and actions to your comments are listed below point-to-point.

To Reviewer #3

Comments 1: Some English expressions should be removed, like ''improvement of living standards'', ''the researchers found that'', ''Fenton reaction is as reaction 1''. Generally the authors should scrutinize whole the text for the English language and write the text to a more formal (scientific) way.

Our Reply: Thank you for your useful suggestion, and we have revised or removed above expressions and revised the manuscript.

Changes: Please see line 27-28.

Comments 2: It should be better, if the authors add more Figures, i.e in the synthetic techniques. Also the addition of one or two tables could help the readers to take more information about the field.

Our Reply: Thank you for your useful suggestion, and we have added a table in the synthetic techniques.

Changes: Please see table 1.

Comments 3: The introduction must be written more extensively and the authors include more information. I think that it is too shorten for review manuscript.

Our Reply: Thank you for your useful suggestion, and we have expanded the introduction, especially in the field of Fenton and MIL MOFs.

Changes: Please see the introduction.

Comments 4: Paragraph 4.2: Ligand. Are there more ligands except from NH2? the authors should mention, and if not why?

Our Reply: Thank you for your useful suggestion, and we have mentioned other ligands in section 4.2.

Changes: Please see section 4.2.

Comments 5: Paragraph 5: Applications. The authors should mention more information about electro-Fenton and photo-fenton, because this is the 'core' of the review.

Our Reply: Thank you for your useful suggestion, and we have mentioned more information about electro-Fenton in section 5.1 and photo-Fenton in section 5.2.

Changes: Please see section 5.1 and 5.2.

Comments 6: It would be better if it is not mentioned so many times, the expression. ''our research group'', one time is enough.

Our Reply: Thank you for your useful suggestion, and we have modified the expression. ''our research group'' in this article.

Changes: Please see line 359 and 454.

Reviewer 2 Report

Authors report the progress in synthesis of Metal organic frameworks materials and their application in the degradation of organic pollutants via Fenton reaction. In this review, the authors have analyzes and summarizes the progress of MILs in degradation of organic pollutant by Fenton reaction in recent years. The analysis is in-depth, and the significance of the work is well stated in the Introduction. However, the author should clearly define the originally of this review. The Introduction section is to short does not provide the current state-of-the-art and their contribution beyond it. The review is not structured in such a way that novelty of research in this area is highlighted at all. So I could not recommend its publication in the present form and much revision are required.

Author Response

Responses to Reviewers’ Comments

Dear Editors ad Reviewers:

Thank you for your letter and for the reviewer’s comments concerning our manuscript entitled “Recent process of MIL MOF materials in degradation of organic pollutants by Fenton reaction”. The authors gratefully acknowledge the reviewers for your helpful comments. We have carefully revised our manuscript according to your comments and remarks. Our responses and actions to your comments are listed below point-to-point.

To Reviewer #2

Reviewer #2 : Authors report the progress in synthesis of Metal organic frameworks materials and their application in the degradation of organic pollutants via Fenton reaction. In this review, the authors have analyzes and summarizes the progress of MILs in degradation of organic pollutant by Fenton reaction in recent years. The analysis is in-depth, and the significance of the work is well stated in the Introduction. However, the author should clearly define the originally of this review. The Introduction section is to short does not provide the current state-of-the-art and their contribution beyond it. The review is not structured in such a way that novelty of research in this area is highlighted at all. So I could not recommend its publication in the present form and much revision are required.

Our Reply: Thank you for your valuable suggestion, and we have expanded the introduction, especially in the field of Fenton and MILs. The development and application of MIL materials have been summarized before, but no one has made a summary under Fenton system, which is a crucial part of advanced oxidation technology. Many degradation processes of MILs are carried out in Fenton system, so the review is meaningful and we also expand the content in the whole manuscript.

Reviewer 3 Report

Manuscript No: catalysts-2240244

Title: Recent progress of MIL MOF materials in degradation of or-ganic pollutant by Fenton reaction.

Comments

Major revision

1.      Section 2.1 and 2.2, heading should be start with capitalize word

2.      Section 4.1 and 4.1.1, headings are not appropriate, should be modified

3.      Characterizations of various MIL MOFS should be discussed in details and various parameters should be compared in table form. Pictorial presentation of characterizations like SEM, TEM, XRD, BET etc should be discussed for various kinds of MIL.

4.      Very little literature is compared in table 1. Lot of literature available on MIL and Fenton reaction as authors mentioned in Fig1. More data should be compared in table 1

5.      1st sentence of Summary and outlook section need revision.

6.      Stability of various MIL should be compared

7.      In introduction, cite these latest articles on MOFs for the removal of pollutants from water like Inorganic Chemistry Communications 145 (2022) 110008, Optical Materials 126 (2022) 112199, Chemical Physics Letters 805 (2022) 139939, Applied Catalysis B: Environmental 268 (2020) 118570.

8.      There are so many typo grammatical errors in whole manuscript, should be revised by some native speaker and formatting should be checked.

Author Response

Responses to Reviewers’ Comments

Dear Editors ad Reviewers:

Thank you for your letter and for the reviewer’s comments concerning our manuscript entitled “Recent process of MIL MOF materials in degradation of organic pollutants by Fenton reaction”. The authors gratefully acknowledge the reviewers for your helpful comments. We have carefully revised our manuscript according to your comments and remarks. Our responses and actions to your comments are listed below point-to-point.

To Reviewer #1

Comments 1: Section 2.1 and 2.2, heading should be start with capitalize word.

Our Reply: Thank you for your useful suggestion, and we have modified section 2.1 and 2.2.

Changes: Please see section 2.1 and 2.2.

Comments 2: Section 4.1 and 4.1.1, headings are not appropriate, should be modified.

Our Reply: Thank you for your useful suggestion, and we have modified section 4.1 and 4.1.1.

Changes: Please see section 4.1 and 4.1.1.

Comments 3: Characterizations of various MIL MOFs should be discussed in details and various parameters should be compared in table form. Pictorial presentation of characterizations like SEM, TEM, XRD, BET etc should be discussed for various kinds of MIL.

Our Reply: Thank you for your useful suggestion, and we have added two tables to compared characterizations of various MIL MOFs and pictorial presentation of characterizations.

Changes: Please see table 1, table 2 and section 2.3.

Comments 4: Very little literature is compared in table 1. Lot of literature available on MIL and Fenton reaction as authors mentioned in Fig1. More data should be compared in table 1.

Our Reply: Thank you for your useful suggestion, and we have added more data in table 3.

Changes: Please see table 3.

Comments 5: 1st sentence of Summary and outlook section need revision.

Our Reply: Thank you for your useful suggestion, and we have revised the 1st sentence of Summary and outlook section.

Changes: Please see line 533-534.

Comments 6: Stability of various MIL should be compared.

Our Reply: Thank you for your useful suggestion, and we have compared the stability of various MIL in section 2.3.

Changes: Please see section 2.3.

Comments 7: In introduction, cite these latest articles on MOFs for the removal of pollutants from water like Inorganic Chemistry Communications 145 (2022) 110008, Optical Materials 126 (2022) 112199, Chemical Physics Letters 805 (2022) 139939, Applied Catalysis B: Environmental 268 (2020) 118570.

Our Reply: Thank you for your useful suggestion, and we have cited Chemistry Communications 145 (2022) 110008, Optical Materials 126 (2022) 112199 in this article.

Changes: Please see Reference 2 and 5.

Comments 8: There are so many typo grammatical errors in whole manuscript, should be revised by some native speaker and formatting should be checked.

Our Reply: Thank you for your useful suggestion, and we have revised the grammar of the whole manuscript.

Round 2

Reviewer 2 Report

Drear editor,

The authors have provide response to most of my suggestions and comments. The anuscript can now be accepted. 

Best regards

Reviewer 3 Report

Accept